# Study of Motor Competence in 4–5-Year-Old Preschool Children: Are There Differences among Public and Private Schools?

**DOI:** 10.3390/children8050340

**Published:** 2021-04-26

**Authors:** Marcos Mecías-Calvo, Carlos Lago-Fuentes, Víctor Arufe-Giráldez, Rubén Navarro-Patón

**Affiliations:** 1Facultad de Ciencias de la Salud, Universidad Europea del Atlántico, 39011 Santander, Spain; marcos.mecias@uneatlantico.es; 2Centro de Investigación y Tecnología Industrial de Cantabria (CITICAN), 39011 Santander, Spain; 3Facultad de Ciencias de la Educación, Universidad de A Coruña, 15008 A Coruña, Spain; v.arufe@udc.es; 4Facultad de Formación del Profesorado, Universidade de Santiago de Compostela, 27001 Lugo, Spain; ruben.navarro.paton@usc.es

**Keywords:** preschool children, Movement Assessment Battery for Children-2 (MABC-2), gender differences, public school, private school

## Abstract

The objectives of this study were (1) to investigate the presence of preschool children with severe motor difficulties (SMDs) and (2) to evaluate the existence of differences in the motor competence (MC) of preschool children from public and private schools based on gender. A total of 581 preschool children (4.66 ± 0.52 years) from Galicia (Spain) were assessed. The Movement Assessment Battery for Children-2 (MABC-2) was used to collect the data. Preschool children from public schools presented a greater number of SMDs (OR = 20.65; CI = 9.99–85.53; *p* < 0.001). Preschool children from private schools have higher scores on the variables studied (for example, manual dexterity (*p* < 0.001), aiming and catching (*p* < 0.001), balance (*p* < 0.001), total test score (*p* < 0.001), and total percentage score (*p* < 0.001)). Regarding the gender factor, differences were found in manual dexterity (*p* < 0.001), aiming and catching (*p =* 0.014), balance (*p* < 0.001), total test score (*p* < 0.001), total percentage score (*p* < 0.001), and in the interaction of both factors in balance (*p* < 0.001), total global score (*p =* 0.004), and total percentage score (*p* < 0.001). Preschool children from private schools are less likely to have severe movement problems and score higher on all dimensions of the MABC-2 than preschool children from public schools analyzed in this study. Likewise, girls score higher than boys in all dimensions, except in aiming and catching.

## 1. Introduction

Motor competence (MC) is a global term that is used to refer to the degree of competence of an individual in the performance of a wide range of motor skills [1], as well as the mechanisms underlying this performance, such as motor control and coordination [2]. In this sense, motor skills are classified in the scientific literature as (1) locomotor skills (e.g., running, sliding, jumping), (2) object manipulation and control skills (e.g., hitting, kicking, throwing, catching), and (3) stability and body control skills (e.g., balance, body rocking) [3]. MC is related to the development of physical and social aspects, as well as the adoption of an active and healthy lifestyle [4,5,6], the delay of which can have long-lasting negative effects [7].

Preschool age is considered a stage of special relevance for the development of MC [8,9] and can be developed through continuous interaction with the social and physical environment [10,11] or overstructured learning environments [12,13]. Therefore, the exposure of preschool children to a certain environment is a criterion that can influence the development of their MC [14], among others. Moreover, many preschool children have motor delays, and during this stage, a large number of young children do not adequately develop this competence [15], as almost one in five schoolchildren are at risk of suffering severe motor difficulties in Spain [16].

With regard to this, schoolchildren in Spain spend a large part of their day in schools (a daily average of 5 h) [17], so these can be considered the fundamental environment for the development of the child [14,18]. Along the same lines, Bronfenbrenner [19], in his bioecological model, proposes that the development of a person is influenced by intrinsic factors to them, such as gender, and extrinsic factors or social environments, such as family, school (microsystem), and friends (exosystem and macrosystem). In this sense, children’s motor development is influenced by the environmental context (i.e., school) and by sociodemographic factors (i.e., gender) [20]. Along the same lines, Newell [21] establishes that the development of MC during childhood depends on biological and environmental factors, as well as the relationships between them. Taking both criteria into account, it seems to be expected that exposure to environments rich in structure and services allows the optimal development of MC [22,23]. In fact, the private school environment, due to its own configuration and organizational independence, seems to positively influence children’s MC [24,25,26]. In Spain, public schools are financed, and their administration depends on the state, so it is up to the educational administrations to provide the public schools with the necessary material and human resources [27]. Furthermore, private schools are educational enterprises; in other words, they are financed exclusively with the quota assigned to families [28]. These types of schools tend to have better facilities or a greater sports offer, in short, greater opportunities to practice [29]. In 2020, 70.06% of preschool children were enrolled in public schools, while 29.04% were in private schools [30]. This is because access to private schools requires high purchasing power and public schools are free [29].

Regarding gender, there are already differences in MC, in this educational stage, in favor of boys, in gross skills [31,32,33,34,35,36,37,38,39] and in fine skills [31,33,34,35,39,40,41,42] and balance [33,34,35,36,41] in girls. Therefore, taking into account what has been previously exposed, it is worth asking whether there are differences in MC between preschool children from public and private schools.

Because, traditionally, MC studies focus mainly on aspects such as age and gender, and there are almost no studies that evaluate the influence of the type of school (public or private) on MC [14,18,43], the objectives of this study were (1) to investigate the presence of students with severe motor difficulties (SMD) and (2) to evaluate the existence of differences in motor competence (MC) of preschool children from public and private schools based on gender in Galicia (Spain).

## 2. Materials and Methods

### 2.1. Study Design

A nonexperimental descriptive cross-sectional design was carried out [44]. The variables of the Movement Assessment Battery for Children-2 (MABC-2) were the dependent variables, comparing them according to the type of school (public/private) and gender (boy/girl).

The research was approved by the Ethics Committee of the national platform Educa (Code 22019) and developed under the standards established in the Declaration of Helsinki.

### 2.2. Participants

The study was carried out in Galicia (Spain) with a nonprobabilistic sample belonging to eight schools selected for geographical proximity and ease of access to the sample (4 public and 4 private).

The inclusion criteria to participate in the study were: (1) provide informed consent signed by their parents or legal guardians; (2) not suffer from illness or difficulty (physical or mental) that prevents participation in the development of the MABC-2 tests; (3) complete the entire process; (4) not having a final score below the 5th percentile after the MABC-2 test, as below this percentile, children can have motor competence problems, thereby altering the results.

A total of 616 preschool children aged 4 to 5 were invited, of which 20 were excluded for not providing the informed consent signed by their parents or legal guardians and 15 for not completing the entire process. Finally, the sample consisted of 581 preschool children. A total of 58 students below the 5th percentile were eliminated from the MC analysis to assess possible differences between public and private schools. These students were used to calculate the risk according to the school but excluded for the comparison of the variables of the MABC-2 battery. Finally, for the comparisons by type of school and gender, the sample consisted of 523 preschool children.

### 2.3. Measurements

The Spanish version of MABC-2 [45] was used. It is a valid and reliable test to identify changes in motor competence in preschool children [45,46,47,48] with very high inter-rater reliability [49].

This battery consists of a standardized test used to identify and describe children’s motor function. For this, it is necessary to carry out a group of motor tests clustered in three dimensions (manual dexterity, aiming and catching, and balance), the duration of which lasts between 20 and 40 min, depending on the age of the child and the degree of difficulty experienced. For the three dimensions of the test and for the total score, the standard and percentile scores are provided as a function of age. The order of application of the tests can be seen in Table 1.

### 2.4. Procedures

The administrations of the schools were contacted to explain to them the objectives of the study. Once the school agreed to carry out the research, the same procedure was conducted with the teachers of the different groups of preschool children. Subsequently, a study information sheet and informed consent were sent to the parents and/or legal guardians of the preschool children to participate. Once accepted, the data were collected.

For the correct evaluation of each test and to try to avoid bias, the evaluators were informed and trained following the general rules of application of the MABC-2 battery manual, recording only the quantitative data in the evaluator’s booklet, without taking into account qualitative data.

To explain each test, the evaluators always performed the same procedure: (1) asking children for verbal consent; (2) description of the task; (3) examiner demonstration; (4) the child practices following the procedure (where the examiner could correct possible errors); and (5) run the test following the instructions in the manual (no instructions were given during the test). In addition, each child was individually evaluated in an isolated, bright, unobstructed, well-ventilated, and noise-insulated classroom provided by the school.

Once the total percentile was obtained, the preschool children were classified according to their equivalence with the MABC-2 battery traffic light system (that is, green: ≥16th percentile: no motor difficulties; yellow: between the 6th and 15th percentiles, both included: in risk of motor difficulties; and red: ≤5th percentile: severe motor difficulties).

### 2.5. Statistical Analysis

For the statistical analysis, the IBM SPSS version 25 software (SPSS v25, IBM Corporation, New York, NY, USA) was used and the level of significance was set at *p <* 0.05. The characteristics of the groups of preschool children are presented in frequency tables as a percentage of categorical data (i.e., gender) and with median values (i.e., age). The chi-square test was used to compare the differences among groups (public vs. private schools) and the color red (children below the 5th percentile) and for the associations the odds ratio (OR) with confidence intervals (CI) of 95%.

The homogeneity of variance was calculated: manual dexterity (*p* = 0.071); aiming and catching (*p* = 0.460); balance (*p* = 0.466); total test score (0.068); total percentile score (*p* < 0.001). The differences in all the variables of the MABC-2 battery between the type of school (public vs. private) and gender (boy/girl) were evaluated using a multivariate analysis of variance (MANOVA), which is a robust enough test to be applied even if the equality of variances is not fulfilled. The effect size was calculated using eta squared (η^2^^2^) and the interaction between variables using the Bonferroni statistic to determine the significance.

## 3. Results

A total of 581 preschool children were evaluated, of which 58 (10%) reached a percentage lower than 5, of which 2 (0.3%) belonged to the group of private schools and 56 (9.6%) to public schools. The odds ratio of SMD was 20.65 (CI = 9.99–85.53; *p* < 0.001) among the children in public schools when compared with the children in private schools.

### 3.1. Baseline Characteristics

The basic characteristics of the 523 preschool children whose total percentile was above the fifth of the MABC-2 battery are shown in Table 2. Of these, 301 (57.6%) belonged to public schools and 222 (42.4) to private schools. A total of 269 (51.4%) were boys and 254 (48.6%) girls.

The baseline results indicate that there are no significant differences regarding the age of the participants (*p =* 0.065), nor the weight (*p =* 0.662), nor the height (*p =* 0.315) but there are in the BMI (*p* < 0.001).

### 3.2. Outcomes According to Type of School and Gender

The results of the multivariate analysis (MANOVA; Figure 1 and Figure 2), regarding manual dexterity, indicate that there is a significant main effect in the type of school factor (F (1, 519) = 93.420, *p* < 0.001, η^2^2 = 0.15), higher in preschool children in private school. A significant main effect was also found in the gender factor (F (1, 519) = 36.262, *p* < 0.001, η^2^2 = 0.06), being higher in girls both in public and private schools with respect to boys but not in the interaction of both factors (*p =* 0.652).

The dimension of aiming and catching shows that there is a significant main effect in the type of school factor (F (1, 519) = 79.200, *p* < 0.001, η^2^2 = 0.13), which is greater in preschool children from private schools, and in the gender factor (F (1, 519) = 6.090, *p =* 0.014, η^2^2 = 0.01), being higher in boys only among boys and girls in private schools (*p* = 0.032). No statistically significant difference was found in the interaction of both factors (*p =* 0.432).

Regarding balance, in the same line as in the previous dimensions, the results indicate that there is a significant main effect in the type of school factor (F (1, 519) = 169.268, *p* < 0.001, η^2^2 = 0.25), being higher in preschool children from private school. A significant main effect was also found in the gender factor (F (1, 519) = 16.264, *p* < 0.001, η^2^2 = 0.03), being higher in girls in both public and private schools compared to boys. In addition, statistically significant results were found in the interaction of both factors (F (1, 519) = 14.629, *p* < 0.001, η^2^2 = 0.03), these differences being between boys and girls in public schools.

The results of the total standard score indicate that there is a significant main effect in the type of school factor (F (1, 519) = 99.706, *p* < 0.001, η^2^2 = 0.16), being higher in preschool children from private schools. A significant main effect was also found in the gender factor (F (1, 519) = 9.142, *p =* 0.003, η^2^2 = 0.02), being higher in girls in public and private schools compared to boys. Furthermore, statistically significant results were found in the interaction of both factors (F (1, 519) = 5.271, *p =* 0.022, η^2^2 = 0.01), these being differences between boys and girls in public schools, with higher scores in girls (*p* < 0.001).

Finally, with regard to the total percentile, the results show that there is a significant main effect in the type of school factor (F (1, 519) = 322.125, *p* < 0.001, η^2^2 = 0.38), being higher in the preschool children from private school. A significant main effect was also found in the gender factor (F (1, 519) = 20.167, *p =* 0.003, η^2^2 = 0.04), being higher in girls. In addition, statistically significant results were found in the interaction of both factors (F (1, 519) = 6.337, *p =* 0.012, η^2^2 = 0.01), as it happened in the total test score, between boys and girls of the public schools, with the scores of the latter being higher (*p* < 0.001).

## 4. Discussion

The objectives of this study were (1) to investigate the presence of students with severe motor difficulties (SMD) and (2) to evaluate the existence of differences in motor competence (MC) of preschool children from public and private schools, based on gender, in Galicia (Spain). The results indicated that the presence of preschool children with SMD was much higher in public schools (9.6%) than in private schools (0.3%). It should be mentioned that this finding is not primarily intended to indicate that the private education system is better than the public one but rather that private schools, due to high market competition, must offer high-quality policies and practices to attract more children to their schools [26].

With regard to the results depending on the type of school, our findings show the existence of a clear difference in the total percentile, with the MC being higher in preschool children from private schools. These results follow the line of previous studies that have also reported these differences [18,24,25,26]. Our findings may make us think that, among other factors, private schools provide more practice possibilities and better facilities for the development of MC [18], as they can provide greater availability and access to play equipment and participation in extracurricular sports classes [26,50,51].

The results were also better in manual dexterity, aiming and catching [18], balance, and the total score, in addition to the total percentile, which confirms that, as in previous studies [8,11,52], MC is the result of the interaction of the individual and the environment in which they are found [19,20]. These results could indicate that the comparison of MC between groups of children from public and private schools suggests that different teaching conditions seem to provide different motor development among children [26]. These different conditions could be the availability of wide and diversified spaces for the practice of physical and sports activities, among others [24], as private schools tend to have a greater number of sports facilities and to offer more extracurricular activities than public ones. The results of this study could also reflect other extrinsic factors, such as differences in family socioeconomics or motor development opportunities at home. These broad and diversified environments allow preschool children to engage in activities that involve large muscle groups that help to develop fundamental motor skills [18].

The results show intergroup differences in schools (public–private) according to gender. Both boys and girls in private schools scored higher in all dimensions of the MABC-2 battery compared to their peers in public schools; that is, manual dexterity was greater than in the study by Barros et al. [53]. Regarding aiming and catching, the results were also higher than in the studies by da Rocha et al. [18] in balance [26] and in global motor performance [18,26]. However, these aforementioned studies did not find differences in any of the motor dimensions among girls, findings that are contrary to the results of our research. These intergroup differences are opposed to the biological factor proposed by Bronfenbrenner [19] and Henderson and Sugden [20], as, biologically, there should be no differences between the same gender; however, the culture in which the children participate [14] and the environment that surrounds them [24,25,26] can modulate MC development. Therefore, the differences found in our study could be the result of better conditions such as facilities, places for free play, and a greater number of extracurricular activities in which the preschool children are involved [54].

In the intragroup analysis by gender in each type of school, both in public and private schools, girls obtained better scores than boys in manual dexterity. These results could be explained and related to the girls’ motivation to perform fine motor skills [55] and writing [32]. Concerning aiming and catching, we only found differences between boys and girls in private schools. This may be due to the fact that the gross motor development of preschool children is associated with the level of stimulation of the school environment [26], which requires the experimentation of activities that use equipment, materials, and adequate instruction; that is, it could be assumed that private schools give more access to these conditions [18], and boys are those who most take advantage of these practice opportunities, which coincides with the results found by Cattuzzo et al. [56] and Spessato et al. [57]. In the balance analysis, only girls in public schools scored higher than boys, which could be explained because girls may have an advantage in terms of developing postural control [33,58]. These results could be due to the fact that the development of motor competence during infancy, childhood, and adolescence depends on biological factors (i.e., genetic, sexual, and maturational) and environmental (i.e., gender roles, experiences, motor play opportunities (variety of play materials and appropriate physical spaces)) [3,14,21,43], and their interactions [21]. Finally, in the total score and the total percentile, differences were only found among boys and girls in public schools, similar to previous research [33,34]. These differences could be partially explained by the higher scores obtained by girls in manual dexterity and balance [33,34,41].

Based on the results obtained in our investigation and due to the small effect size, care must be taken in interpreting the results, as it has not been possible to report the causality between the variables studied due to the cross-sectional nature of the study. However, these results may stimulate more studies to try to establish a causal relationship between different levels of the Bronfenbrenner model and MC. Therefore, more research is needed in this educational context (public vs. private centers) to understand the cause and effect of some factors (i.e., contextual, family, or individual) on others (i.e., MC) in these age groups.

In addition to the contributions of this study, certain limitations must be taken into account that should make the results viewed with caution. Extracurricular sports practice has not been considered, neither the lack of information on the number of hours children play outdoors (not only at school but also with their families) nor the frequency and schedule of Physical Education classes. Another important factor that can influence these results is the family environment, such as economic situation, education level of parents, among others [19]. Likewise, the design of this study does not allow conclusions to be drawn about the causality of the results found. On the other hand, it must also be taken into account that the MABC-2 battery does not discriminate between good or excellent motor competence scores.

## 5. Conclusions

The results obtained in this study allow us to conclude that, in this sample, predominantly more children from public preschools have severe motor difficulties compared to children from private preschools.

Preschool children from private schools obtained higher scores in manual dexterity, aiming and catching, and balance, as well as the total score of the eight-battery test and the total percentile. In between-group analyses, differences were found for boys but not for girls.

In gender analysis, girls score higher than boys on all dimensions (i.e., manual dexterity, balance, total score, and total percentile) but not on the object control dimension (aiming and catching). Girls from private schools show better scores in manual dexterity and boys in aiming and catching, with similar performance in the rest of the scores. In public schools, girls score better in all dimensions than boys, except aiming and catching.

## 6. Practical Applications

Based on the results obtained, and taking into account that private schools are supported by the income from their activities and usually offer greater possibilities of practice, we suggest an adjustment in the opportunities for the practice and creation of suitable environments for play and practice in the different educational centers (both public and private). We also suggest that teachers tailor their programming to the motor development of preschool children and be more short-term. Finally, we suggest that the promotion of practice opportunities and exposure to suitable, spacious environments with the availability of materials are essential factors for the equitable development of MC in children from public and private schools.

## Figures and Tables

**Figure 1 children-08-00340-f001:**
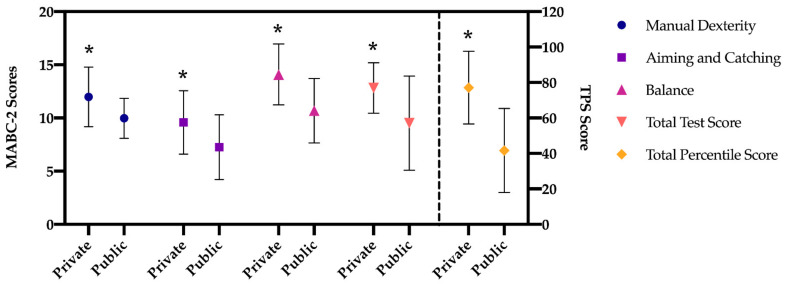
Differences in MABC-2 scores between private and public schools. Note: scores (range: 1–19); TPS score (range: 0.01–99.9); * *p <* 0.001.

**Figure 2 children-08-00340-f002:**
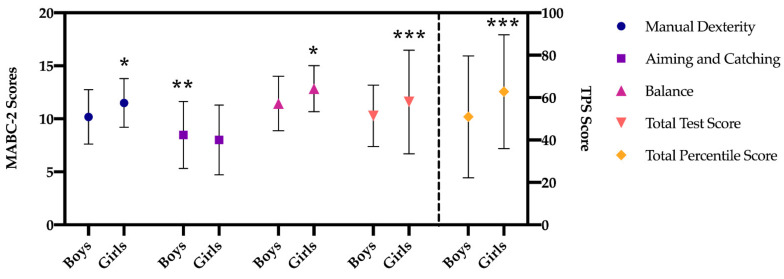
Differences in MABC-2 scores between boys and girls of the total sample. Note: scores (range: 1–19); TPS score (range: 0.01–99.9); * *p <* 0.001; ** *p <* 0.05; *** *p <* 0.005.

**Table 1 children-08-00340-t001:** MABC-2 test [45,46].

Test Dimension (Range)	Subtest
**Manual dexterity**	1st: Post coins (MD1)
	2nd: Threading beads (MD2)
	3rd: Drawing trail (MD3)
**Aiming and catching**	4th: Catching bean bag (AC1)
	5th: Throwing bean bag onto mat (AC2)
**Balance**	6th: One-leg balance (Bal1)
	7th: Walking heels raised (Bal2)
	8th: Jumping on mats (Bal3)
**Total test Score (1–19)**	MD1 + MD2 + MD3 + AC1 + AC2 + Bal1 + Bal2 + Bal3
**Total** **percentile Score (0.1–99.9)**	

Note: MD: Manual dexterity; AC: Aiming and catching; Bal: Balance.

**Table 2 children-08-00340-t002:** Baseline characteristics of participants.

	Public School (*n* = 301)			Private School (*n* = 222)		
	All	Boys (*n* = 164)	Girls (*n* = 137)	All	Boys (*n* = 105)	Girls (*n* = 117)
Age (years)	4.6 ± 0.53	4.6 ± 0.55	4.6 ± 0.50	4.7 ± 0.30	4.7 ± 0.31	4.7 ± 0.29
Weight (Kg)	19.1 ± 2.02	19.1 ± 1.88	19.2 ± 2.11	18.9 ± 3.07	19.4 ± 2.38	18.5 ± 3.60
Height (cm)	108.2 ± 0.41	108.7 ± 0.26	108.6 ± 0.47	109.0 ± 0.71	111.0 ± 0.64	108.2 ± 0.76
BMI (kg/m^2^)	19.1 ± 2.02	19.1 ± 1.88	19.2 ± 2.11	17.2 ± 2.49	17.6 ± 1.96	16.8 ± 2.90

Note: data are presented as mean ± standard deviation mean.

## Data Availability

The data presented in this study are not available in accordance with Regulation (EU) of the European Parliament and of the Council 2016/679 of 27 April 2016 regarding the protection of natural persons with regard to the processing of personal data and the free circulation of these data (RGPD).

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
