# Peer review of "Study of Motor Competence in 4–5-Year-Old Preschool Children: Are There Differences among Public and Private Schools?"

_children, 2021, doi:10.3390/children8050340_

Round 1

Reviewer 1 Report

See attachement. 

Reviewer 2 Report

The revised article entitled

Study of motor competence in 4-5 years-old preschool children:

 Are there differences among public and private schools?

The theme is highly interest for researched in the field of early childhood education, focusing specifically on the analysis of motor skills of a group of children, between 4 and 5 years old, preschool age, as an essential factor for a healthy lifestyle.

It presents a abstract  that immediately presents the two objectives of the study, the sample, the research instrument (MABC2), the most relevant results and (lines 13-28) and appropriate keywords. Consider that the objectives are clear (lines 13-15) as well as important variables for the dependent variable - the battery used to assess the quality of movement or motor competence; independent variables - type of school (public / private) and gender (boys and girls); focuses on the most relevant results and presents a brief conclusion and another point with practices sugestions.

Firstly, for the Introduction (lines 32-68), the authors propose an analysis with the most relevant concepts and theories for the study variables (motor competence; development sustained by extrinsic factors and considering Bronfenbrenner's bioecological model) and in-depth referencing about of 36 studies to frame their problem - in Spain, one in five children has severe motor difficulties.

Improvement suggestions: similarly to what was done for Motor Skills, here it introduced an abbreviation for“ Severe Motor Difficulties

The materials and methods (from line 69-146) are very consistent and it is very clear the whole process adopted for a research with characteristics of non-experimental study, which involves a population of children in Spain, but which was carried out according to the expected standards ( 1975 Helsinki Declaration; Informed Consent signed by parents after acceptance by school and educators). They explain all the steps in selecting the sample that has a relevant change. For a recurrence of a battery of tests, adapted to the study population and described as evidence of each of the three dimensions under analysis, as well as a determination of the scores (by dimension, total or total percentage) and the application protocol. For an analysis of the data they resort to the statistics of some power.

Improvement suggestions: The last paragraphs of points 2.3. (from line 105-109) and 2.4. (from line 131-134), about the final classification in the MABC-2 Battery, are repeated. I suggest a simplification in point 2.4., presenting only the values, since in the results the “traffic light system” is no more referred; In point 2.5., It presented the symbol of the “Chi-Square Test” (line 140); and, I have doubts about the reference to “Partial Eta –squared”, because the symbol that is presented throughout the work should be - “Eta –squared” (line 145).

The results, for the analysis of the 1st objective are brief and determinant (lines 148-153) and for the analysis of the 2nd proposed objective, they start from the careful analysis of the data presented in two tables and two graphs, full of interesting information.

Improvement suggestions:  In table 1 - Extend BMI, even if it is in the analysis performed throughout the text (line 161); uniform the decimal places of the values apresented for the averages; in the titles of Figure 1 and 2 – Differences (?) between…. (line 192 and line 195) – I think  something in  missing – for exemple the reference to MABC-2.

In the discussion carried out, he requested many studies of the areas under analysis and that, for this reason, supports his results well.

Improvement suggestions: I have doubts about the concepts “worth” used in line 218 and “Matches” in line 261 are the most appropriate;

The conclusion, in general reflect the results achieved according to the objectives outlined.

Improvement suggestions: Review the phrase “In gender analysis, girls score higher in all dimensions ..” line 281

It presents a point 6 of very interesting practical applications. .

The final references are in their current applicants (about 33 are 65% of publications in the last 11 years, 35% are over 11 years old) and address the major themes covered, most in reference journals. Only 6% of the references mentioned include authors of the article, in very current articles (all from 2021) and related to the theme studied.

O artigo revisto e intitulado

Study of motor competence in 4-5 years-old preschool children: Are there differences among public and private schools?

A temática tem uma elevada relevância para a área da educação infantil, incidindo especificamente na análise da competência motora de um grupo de crianças, entre os 4 e os 5 anos, idade pré-escolar, como fator essencial a um estilo de vida saudável.

Apresenta um resumo que apresenta desde logo os dois objetivos do estudo, a amostra, o instrumento de pesquisa  (MABC2), os resultados mis relevantes e as conclusões (linhas 13-28) e palavras chave apropriadas.

Considero que os objetivos são claros (linhas 13-15) inclui bem as variáveis importantes para a análise   variável dependente -  da bateria utilizada para avaliar a qualidade de movimento ou competência motora; variáveis independentes -  tipo de escola (pública / privada) e género (rapazes e raparigas); foca os resultados mais relevantes e apresenta uma breve conclusão e um ponto 6 com propostas práticas.

Primeiramente, para a Introdução (linhas 32-68), os autores propõem uma análise que enquadra conceitos mais relevantes para as variáveis do estudo (Competência Motora;  desenvolvimento sustentado em fatores extrínsecos e  considerando o modelo bioecológico de Bronfenbrenner) e e aprofundada referenciando cerca de 36 estudos para enquadrar a sua problemática – em Espanha, uma em cada cinco crianças tem dificuldades motoras severas.

Sugestões de melhoria: à semelhança do que foi realizado para a Competência motora introduzia aqui uma abreviatura para “Severe Motor Difficulties “ 

Os materiais e métodos (da linha 69-146) estão muito consistentes e fica bem claro todo o processo adotado para uma pesquisa com características de estudo não experimental, que envolve uma população de crianças de Espanha mas que foi realizado seguindo as normas previstas (Declaração de Helsinque de 1975; consentimentos Informados assinados pelos pais após a aceitação pela escola e educadores). Explicam todas as etapas de seleção da amostra que tem uma dimensão relevante. Para a avaliação recorrem a uma Bateria de Testes, adaptada à população em estudo e descrevem as provas de cada uma das três dimensões em análise, assim como, a determinação dos scores (por dimensão, total ou percentil total) e o protocolo de aplicação. Para a análise dos dados recorrem a técnicas estatísticas de algum poder.

Sugestões de melhoria: Os últimos parágrafos dos pontos 2.3. (da linha 105-109) e 2.4. (da linha 131-134), acerca da classificação final na Bateria MABC-2, repetem-se.  Sugiro uma simplificação no ponto 2.4., apresentando apenas os valores, já que nos resultados o “traffic light system” não é mais referido; No ponto 2.5., apresentava o símbolo do “Chi-Square Test” (lina 140); e, tenho dúvidas quanto à referência ao “Partial Eta –squared” , pois pelo símbolo que é apresentado ao longo do trabalho deverá ser -  “Eta –squared”  (linha 145).

Os resultados, para a análise do 1.º objetivo são breves e determinantes (linhas 148-153)  e para a análise do 2.º objetivo proposto partem da análise cuidada dos dados apresentados em duas tabelas e dois gráficos, repletos de informação interessante..

Sugestões de melhoria:  Na tabela 1 – Colocar por Extenso o BMI, mesmo que seja na análise realizada ao longo do texto (linha 161) ; uniformizar as casas decimais dos valores apresentados para as médias; nos títulos da Figura 1 e 2   - Differences  ?  between ….  (linha 192 e linha 195)  - Acho que falta algo - por exemplo, a referência ao MABC-2.

Na discussão realizada solicita muitos estudos das áreas em análise e que, por isso, sustenta bem os seus resultados.

Sugestões de melhoria: Tenho dúvidas que os conceitos “worth” utilizado na linha 218  e “Matches”  na linha 261 sejam os mais adequados.

As conclusões, refletem em geral os resultados alcançados em função dos objetivos delineados. 

Sugestões de melhoria: Rever “In the gender analysis, girl score higher in all dimensions ..”  linha 281

Apresenta um ponto 6 de aplicações práticas muito interessante.

As referências finais são na sua maioria atuais (cerca de 33 são 65 % de publicações dos últimos 11 anos, 35 % com mais de 11 anos) e abordam as grandes temáticas abordadas, a maioria em revistas de referência. Só 6% das referências mencionadas incluem autores do artigo, em artigos muito atuais (todos de 2021) e relacionados com a temática estudada.

Reviewer 3 Report

The study presented is relevant for this area and therefore authors are to be congratulated.

Despite it is a convenience sample, the number of participants is significant.

It is well written and small corrections will be suggested.

Specifically:

  1. Abstract. Line 16. Substitute participated by were assessed.
  2. Introduction. Line 34. Remove the word This.
  3. Materials and methods

Lines 82-84. Should be mentioned which were inclusion/ exclusion criteria. In the participants were there only children with typical development? Were excluded children neurodevelopmental disorders?

Line 91. Remove the MABC-2 designation in full. Its was already done in line 72.

Lines 98, 103, 104, 128-130, 179. Replace designation scalar score(s) by standard score(s) to stay in agreement with MABC-2 manual.

Line 108. Substitute ambar by yellow.

Line 120. Should start by indicating that if the child was asked for verbal consent.

Line 126. Replace direct scores by raw scores.

Lines132-134. There is no need to repeat traffic light explanation. It was already done in lines 106-109

Line 141. It is not clear what the authors mean by the color red (yes-no).

Line 145-146. Authors should inform which is partial eta squared values reference in order to be clear in results what is considered a small, intermediate or high effect.

  1. Results

Line 158. Table 1 has information about BMI, which appears for the first time without any previous information in introduction.

It is sufficient (and in agreement with MABC-2 authors) that research results can be presented only using standard scores, without the need to also present the percentile scores.

  1. Discussion

It is suggested that authors include has a limitation the lack of information regarding the amount of hours that children play outdoors (not only in school but also with their families).

Round 2

Reviewer 1 Report

Authors have addressed some of the comments, however, I the major concern regarding causality has only been partially resolved.

Because longitudinal follow-up is lacking it is essential to document factors that may influence motor competence (and development from 0 to 4 years) of the two samples. Three obvious examples would be SES, age at which children started education, nature of the context. It is also important to include these matters in the interpretation and discussion of the results. Authors only do so at the very end of the discussion.

The phrases that have been revised to resolve the causality issue (L229, L236, L256) only contain more information on why the authors believe there would be causal link. It is not accepted that the difference may be due to a completely other reason.

  • - L254: Private school provides more practice possibilities and better facilities > Could it be that the motor competence of children attending private schools was greater at the start, in other words that they had better motor development opportunities at home?

  • - L265: Since private schools tend to have greater number of sports facilities and tend to offer more extracurricular activities > A tendency means that not all private schools will provide these facilities/activities, but this would be easy to check in your sample. Also, do we know that children at private schools attend these activities? Do we have an idea of the degree of physical activity outside the PE curriculum of both groups?

  • - L282: Better conditions such as facilities, places for free play… > Has this been checked?

In all these examples Authors link the difference to the context of the schools, and completely neglect potential other causes.

Description of the statistics and the interactions:
L157: Authors refer to frequency tables and median values which have not been reported here. If this refers to the exploratory analysis prior to conducting inferential statistics this can be omitted.

L161: Please add which test was used to check homogeneity of variance

L165: Something I did not notice in my first review. A MANOVA was performed. I presume this analysis only included the scores on three dimensions (manual dext, aiming/catching, balance) and a separate ANOVA was performed on the total score. As the total score is summation of the three dimensions it would be inappropriate to include this in the MANOVA.

L196: If there is not interaction between school type and gender, the main effect of gender indicates that there is a difference between boys and girls in both public and private school. The fact that the p-value of a pairwise comparison indicates a significant difference in private schools only, is probably due to the fact that the size of the main effect is really small.

L 205: This phrase is unclear.

L211-212: This phrase is unclear. A significant interaction effect means that the difference between girls and boys is greater in either private or public schools. Again, given the very small effect size one should ask whether this statistical difference is relevant.
